# Data Augmentation for Facial Recognition with Diffusion Model

Zhiqi Huang[1*]    Hao Geng[1*]    Haoyu Wang[1]    Huixin Xiong[2]    Zhiheng Li[1✉]

[1]Tsinghua University    [2]Megvii

{huangzq22, gengh23, haoyu-wa22}@mails.tsinghua.edu.cn

zhhli@mail.tsinghua.edu.cn    xionghuixin@megvii.com

[*] Equal Contribution    ✉ Corresponding authors

## Abstract

*In recent years, facial recognition technology has made significant progress. However, it also faces challenges in common scenarios of daily life. For example, facial accessories such as masks, glasses, and hats have a negative impact on recognition accuracy. This paper introduces a facial data synthesis pipeline based on the diffusion model, which combines the text-to-image generation method with Mask-ControlNet. The pipeline can generate various common facial occlusions, achieving diverse and high-fidelity facial image generation. By comparing the performance of different models trained with synthetic and real images, extensive experimental results confirm the effectiveness of this method in enhancing the robustness of facial recognition.*

## 1. Introduction

Facial recognition is a technology that matches faces in digital images or videos with faces stored in a database. Currently, facial recognition based on deep convolutional networks such as DeepFace [1], FaceNet [2], and OpenFace [3] are widely used, and they typically achieve high accuracy on normal faces. However, in daily life, factors such as hairstyles, accessories, and clothing may affect recognition performance. The reason is that the collected samples are relatively single and limited in quantity.

To tackle these challenges, researchers have proposed some data augmentation methods. As shown in the first row of Fig 1, they manipulate original images by rotating, flipping, scaling, cropping, adding noise, etc. to generate additional training samples. In addition, some GAN-based methods, such as DiscoGAN [4] and BeautyGAN [5]. These methods can be used for hairstyle transfer and facial makeup transfer. As shown in the second row of Fig 1, diffusion-based generative models like DiFaReli [6] and Diffusionrig [7] have also been utilized for facial image restoration and augmentation.

Before data augmentation, these generative models may

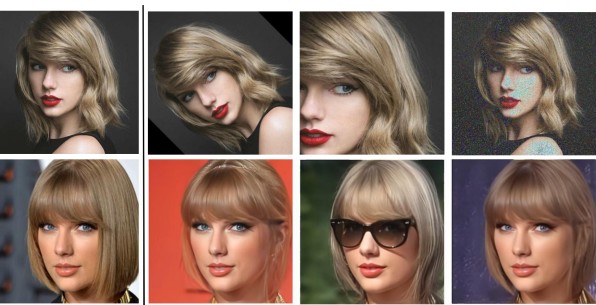

Figure 1. Left column is original data. The right shows the images after data augmentation. The upper row is based on geometric transformations and pixel operations. Lower is showing various transfers using Diffusionrig [7].

need to first obtain prior knowledge of a person's face in order to have a high degree of restoration, such as DreamBooth[8]. Inpainting methods[9], although having high object fidelity without additional training, may result in facial disharmony in the generated images.

Our method solves this problem by introducing an additional mask prompt. We fine-tune the pre-trained diffusion model without prior knowledge of specific faces. With the great advancements of large vision models like SAM [10], the facial mask can be easily obtained. Specifically, the reference image is first fed to SAM to produce a mask to segment the regions of the face. Then, the resultant image is concatenated with the reference image as the conditional information for image synthesis. The additional mask prompt facilitates the network to better maintain the facial details and model the pixel relationship of segmented edges, resulting in higher-quality synthetic images.

The main contributions are summarized as below:

1) We propose a framework termed Mask-ControlNet to achieve higher-quality facial image generation by introducing an additional mask prompt. With the help of this mask prompt, different accessories and hairstyles can be synthesized for facial data, while maintaining high-

fidelity facial features.

2) We conduct extensive comparative experiments to test the data augmentation effect of generated images. Quantitative and qualitative results show that our framework can generate diverse and high-quality datasets, which can effectively improve the robustness of facial recognition.

## 2. Related work

In this section, we introduce some common methods of facial recognition and related datasets. Next, we briefly review recent advances in generative models that can be used for facial data augmentation.

### 2.1. Facial Recognition Models

Over the past few years, significant advancements in deep neural networks, coupled with the employment of extensive facial datasets, have markedly improved the performance of facial recognition systems. State-of-the-art facial recognition models [11–15] use large-scale face datasets such as DeepFace [1], FaceNet [2], and VGGFace2 [16] to optimize the performance [17–22]. To avert overfitting and to enhance the robustness of these systems, the training dataset needs to simulate the diversity and unpredictability of the real world.

### 2.2. Generative Models for synthesizing faces

Due to privacy concerns and resource limitations associated with collecting real facial datasets, there are technical challenges in gathering diverse facial images. Recent research has pivoted towards employing synthetic data as a substitute for real data in the training of facial recognition systems. Major deep generative models, such as Variational Autoencoders (VAE) [23], Generative Adversarial Networks (GAN) [24, 25], Autoregressive Models [26], and Diffusion Models [27–29], are capable of sampling from existing data distributions to generate synthetic data that closely resembles real-world data. Specific GANs [30–32], including CONFIG [33], DiscoGAN [34], VecGAN [35], and Face-ID-GAN [36], can use predefined attributes to regenerate multiple faces of an existing one. In addition, Stable Diffusion [37] has been shown to generate diverse and photorealistic faces, enriching datasets and improving model performance as demonstrated by a recent quantitative analysis [38].

## 3. Methodology

Given a facial image of a certain person, our goal is to generate an image that maintains high fidelity of facial details, while synthesizing different contexts and compositions based on text prompts.

### 3.1. Training-time Framework

As shown in Fig. 2, our framework is built on top of a diffusion model and is trained in a self-supervised manner. First, the input image is fed to the VAE encoder to obtain feature maps $F$ and then the noise is progressively added, resulting in $F_t$. Here, $t$ represents the number of times noise is added. Afterward, the noisy feature maps $F_t$ are passed to the diffusion model to predict the noise and reconstruct the input image.

In parallel to the main path, our framework has an image branch and a text branch to provide additional conditions for the diffusion model. In the image branch, the input image and text prompt are first fed to GroundingDINO [39] to obtain object detection, which is a kind of box prompt used for SAM [10] to produce the object mask. Then, the resultant mask is used to segment the object in the image. Next, the concatenation of the object image and the image is passed to an adapter layer. Afterward, a VAE encoder and ControlNet are employed to control the diffusion model to reconstruct the input image. In the text branch, BLIP[40] is adopted to extract textual descriptions of the input image. Then, the extracted text prompt is fed to CLIP [41] to provide additional control to the diffusion model. Finally, the features extracted from the text and image prompts are connected to the diffusion model with zero convolution layers.

During optimization, only the adapter layer and the ControlNet are trainable while the diffusion model is frozen. The loss function used is defined as:

$$L = E_{z_0, t, c_t, c_f, \epsilon \sim N(0,1)} ||\epsilon - \epsilon_\theta(z_t, t, c_t, c_f)||_2^2, \quad (1)$$

where $z_0$ represents the data in the latent space, $c_t$ and $c_f$ are the text condition and the latent condition, respectively.

### 3.2. Inference-time Framework

When synthesizing facial data for model training, there are two methods. The first is to preserve facial features and change people's clothing and hairstyle. This requires feeding reference images to SAM to produce a mask to segment the face. Another approach is to generate occlusions on the face, which requires the use of the facial keypoint detection method (MTCNN [42]). For example, if we want to generate a face with a mask, we first need to detect the position of the nose, mask the facial area below the center of the nose, and imply in the text prompt that the person in the image is wearing a mask. In addition, sunglasses, forehead bangs, etc. can also be batch generated through this method. The synthesis effect can be shown in Fig. 4. Then, the concatenation of the mask and the segmented face image is passed to the VAE encoder. Meanwhile, the text that describes the context of the generated image is fed to CLIP. Next, the features extracted from the image and the text prompt are

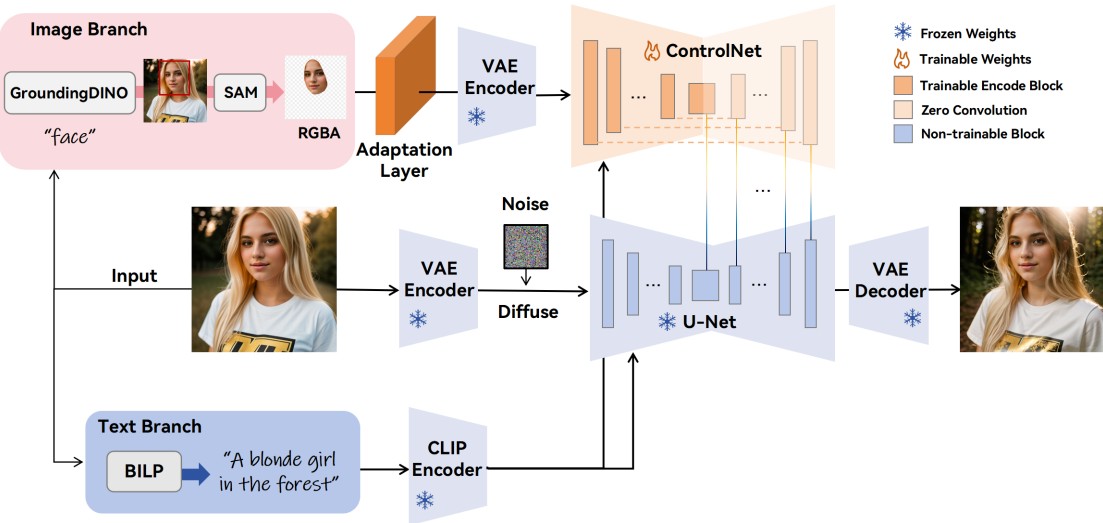

Figure 2. An illustration of our framework during the training phase.

passed through ControlNet and used as conditions for the diffusion model to synthesize an image from a noise image.

# 4. Experiments

In this section, we first introduce the experimental setup and visualize some face data synthesized by Mask-ControlNet. Next, we conduct several comparative studies to test the performance of different models in face recognition tasks, demonstrating the effectiveness of synthesized images in data augmentation.

## 4.1. Experimental Setup

During the training phase, we collected 18,000 images from numerous websites using keywords such as people, cosmetics, art photos, and clothing. In addition, we selected 20,000 images from the SA1B and COCO datasets. Mask types include people, faces, clothing, various accessories, etc. After data cleaning and annotation, a total of 38,000 valid images and approximately 50,000 valid masks are obtained as the training set.

In the face recognition task, our training and testing sets are sourced from the facial recognition dataset collected from Pinterest. To test the robustness of the model, there are a total of 105 people, each has 100-200 facial images with significant differences, making facial recognition challenging.

We perform various types of enhancements on each face in the training set, including changing hairstyles and clothing, adding facial accessories, and randomly generating facial features. The synthetic images generated by Mask-

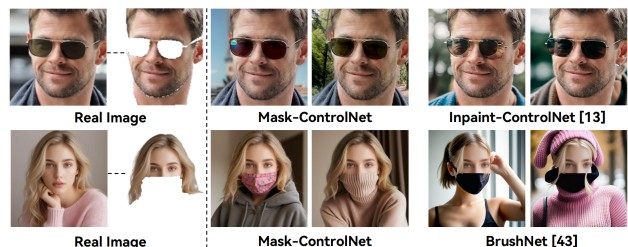

Figure 3. Comparison of synthesized images among Mask-ControlNet, Inpaint-ControlNet [9] and BrushNet [43]

ControlNet were created using 10-20 images per individual. These synthesized training data are used to train the facial recognition model, and then we test the performance of these models using closed set recognition.

## 4.2. Performance Evaluation

### (1) Facial Data Synthesis

As shown in the first row of Fig. 3, compared to Inpaint-ControlNet [9], our method can generate more diverse and coordinated images, but the background generated by Inpaint is relatively single and the glasses are also very uncoordinated. Compared to BrushNet [43], our method can generate more real and restored images. As shown in Fig. 4, we only need to provide the model with a segmented facial image and a text prompt to control the redrawing of the entire image. From the figure, it is evident that within the mask region, a variety of occlusions, clothing, and hairstyles can be realistically synthesized, while in the non-mask area, the key facial features are still high fidelity.

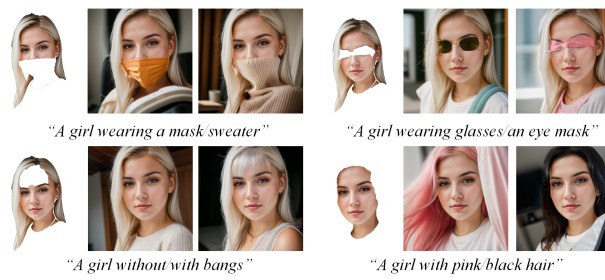

*"A girl wearing a mask/sweater"*      *"A girl wearing glasses/an eye mask"*

*"A girl without/with bangs"*      *"A girl with pink/black hair"*

Figure 4. Four ways to synthesize images. The left of each example serves as the input for Mask ControlNet, while the right contains some images generated under the guidance of the text prompt.

When synthesizing large-scale datasets, the human face can be accurately segmented, but the areas of the eyes, forehead, and mouth are difficult to accurately mask. At present, we only generate glasses, masks, etc. roughly near the key points, which may interfere with training performance. In the following work, we will focus on improving the process of this part to enhance the quality of data synthesis.

**(2) Quantitative Results**

First, we evaluated the performance of two augmentation methods(random and Mask-ControlNet augmentation) on the three models. Specifically, we trained VGG19 [44] and ResNet50 [45] from scratch and fine-tuned ViT [46] (ViT-B). We compared the test metrics across three levels of dataset augmentation, where the expanded datasets were proportionally scaled to 1x, 2x, and 4x the size of the initial dataset.

The outcomes presented in Tab. 1 demonstrate that, across different dataset augmentation ratios, the Mask-ControlNet augmentation (MCA) consistently achieves superior results compared to random augmentation for all three models. As the proportion of the MCA datasets increases, the effectiveness also improves progressively. However, random enhancement methods such as cropping and rotation can easily lead to overfitting during training. Consequently, we have grounds to believe that our MCA plays a beneficial role in expanding the dataset.

Additionally, we compared four representative face recognition methods: FaceNet [2], ResNet50 [45], InceptionResnetV1 [47, 48], and ViT [46]. The experimental data consists of 100×105 real images and 100×105 generated images. Mask ControlNet generates occlusion of random areas on the face by generating accessories, with the ratio of accessories to face area set to 0%, 20%, and 40%, respectively. We trained FaceNet and fine-tuned the other three models and evaluated their recognition accuracy on the same test set. Additionally, we compared the similarity between the feature vectors(ViT-B encoded) of the gener-

Table 1. Comparison of accuracy across different data augmentation methods and dataset enhancement proportions in two models. The ratio in the table denotes the ratio of the original dataset to the augmented dataset. RA indicates data enhancement using the Random Augmentation method, while MCA refers to augmentation employing Mask-ControlNet.

|  | 1:1 | | 1:2 | | 1:4 | |
|---|---|---|---|---|---|---|
|  | RA | MCA | RA | MCA | RA | MCA |
| VGG19 | 0.386 | **0.403** | 0.377 | **0.466** | 0.453 | **0.483** |
| ResNet50 | 0.404 | **0.592** | 0.519 | **0.622** | 0.511 | **0.656** |
| ViT | 0.875 | **0.886** | 0.890 | **0.912** | 0.914 | **0.929** |

Table 2. Comparison of feature similarity and accuracy on four models for synthetic data with different proportions of mask. IRV1 in the table represents InceptionResnetV1. 0% of the mask indicates that the face in the original image has been regenerated without any changes.

| Mask | Similarity | FaceNet↑ | ResNet↑ | IRV1↑ | ViT↑ |
|---|---|---|---|---|---|
| 0% | 68.94 | **0.257** | 0.947 | 0.865 | 0.887 |
| 20% | 65.40 | 0.251 | **0.958** | **0.903** | **0.931** |
| 40% | 62.25 | 0.225 | 0.953 | 0.856 | 0.906 |

ated images and the original images.

As shown in Tab. 2, as the area of the mask increases, the similarity between images gradually decreases, but the difference is not significant, indicating that the synthesized image can still retain most features of the face. Among the four methods, except for simple networks like FaceNet(CNN) which perform slightly worse than the original data (0.251 vs 0.257), all other methods perform best on synthetic faces with a mask of 20%. This indicates that masking a smaller portion of the face is of great help in supplementing the dataset, while masking a larger area of the face may introduce some noise, thereby interfering with the training of the model.

## 5. Conclusion

In this paper, we present a simple yet effective framework to synthesize high-quality facial images with an additional mask prompt. With this conditional information, the network can well capture the relationship between object edge pixels. From the quality of generated images, our method can synthesize real and high-fidelity facial images, including various facial occlusions, clothing, and hairstyles. Extensive experiments demonstrate the effectiveness of our synthesized data. Additionally, we expect to explore how to more accurately recognize different regions of the face and achieve more efficient and high-quality facial data synthesis in the future.

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
