# OpenReview forum: "Data Augmentation for Facial Recognition with Diffusion Model"
_thecvf.com/CVPR/2024/Workshop/SyntaGen — SyntaGen 2024_

### Official Review · Reviewer_fRJb · 2024-03-31
**Review of the paper, decision: Reject.**

**Rating:** 4
**Confidence:** 3

**Review:**

The paper provides a method to further fine-tune a Stable Diffusion model, accompanied by the auxiliary model to control the augmentation on the face. The overall pipeline is reasonable, albeit not completely novel, and the qualitative results are quite promising. However, the quantitative results are not satisfactory. The authors provide two evaluations: (1) cross-comparison in the accuracy of the face recognition models trained with Random Augmentation and theirs and (2) an ablation of mask size effect upon face similarity by multiple recognizers. Table (1) only shows a relative comparison with a simple augmentation method, rather than comparing with the common face benchmark dataset, also the accuracy metric is not a reasonable metrics due to its bias w.r.t the positive and negative samples. On the other hand, (2) doesn't show which portion of the face is masked w.r.t the amount of masking percentage, which are highly vital to explain the similarity degradation. Hence, I would like to see a more elaborated evaluation, for now, I keep my review score to be reject.

---

### Official Review · Reviewer_w64K · 2024-04-01
**The paper is satisfactory, yet it could benefit from further improvements.**

**Rating:** 6
**Confidence:** 4

**Review:**

Summary:
This paper aims to enhance facial recognition tasks by augmenting per-person training images with synthetic data. This setup is challenged because the generated images must contain high variation (e.g., poses, wearing glasses, wearing accessories) while preserving the person's identity. To achieve this, the work proposes conditioning the pretrained diffusion model with an object mask created by SAM and an additional Adaptation Layer. The object mask is then fed to the pretrained diffusion model via ControlNet. The paper demonstrates that both CNN-based and transformer-based face recognition models trained using the proposed augmentation method surpass those trained with traditional augmentation methods (e.g., geometric translation, pixel operation).

Pros:
1. The paper effectively motivates the problem.
2. The paper solves the challenging task of data synthesis that need to ensure that the generated data preserves the subject's identity.

Cons:
1. Lack of qualitative results. Although the paper mentions augmentation with accessories, only the results for sunglasses are shown.
2. While diffusion inpainting seems to be the main point of comparison, given the improvements might stem from the pretrained diffusion model's ability to generate varied images, only one subject/mask comparison to diffusion inpainting is shown.

---

### Official Review · Reviewer_41jF · 2024-04-03

**Rating:** 6
**Confidence:** 4

**Review:**

This paper presents a pipeline for augmenting facial images for recognition. In contrast to previous methods that utilize basic augmentations such as rotation, this paper focuses on more semantically meaningful augmentations, such as adding masks and changing hair color. The pipeline is based on the mask inpainting model of the diffusion model. With this augmentation technique, it does not exhibit saturation when the amount of synthetic data is increased. The advantage lies in its potential to enhance data variation for face recognition. However, the drawback lies in some grammatical errors (for example, line 130).

---

### Decision · Program_Chairs · 2024-04-06

**Decision:**

Accept

**Comment:**

The paper received mixed reviews, with 2 Borderline Accept and 1 Reject scores. The reviewers applauded the paper's novel augmentation technique that focuses on more semantically meaningful augmentations, e.g., adding sunglasses or changing hair color, via a diffusion-based mask inpainting model. However, there are severalconcerns on the presented experiments raised by the reviewers: (1) the face recognition protocol and evaluation metric are non-standard, (2) while different face masking regions may have different effects on the face recognition performance, the authors did not elaborate on which portion of the face was masked, (3) ablation studies to confirm that the improvements mainly came from the diffusion inpainting augmentation is needed.

Despite the mentioned shortcomings, the ACs found the proposed augmentation technique interesting and valuable for the community. The submission is also for a short paper only. Hence, we decided to accept the paper. The authors should improve the experiment part to address the mentioned weaknesses.